# ADVERSARIAL TEXT TO CONTINUOUS IMAGE GENERATION

## ABSTRACT

Implicit Neural Representations (INR) provide a natural way to parametrize images as a continuous signal, using an MLP that predicts the RGB color at an $(x, y)$ image location. Recently, it has been shown that high-quality INR decoders can be designed and integrated with Generative Adversarial Networks (GANs) to facilitate unconditional continuous image generation that is no longer bound to a particular spatial resolution. In this paper, we introduce HyperCGAN, a conceptually simple approach for Adversarial Text to Continuous Image Generation based on HyperNetworks, which produces parameters for another network. HyperCGAN utilizes HyperNetworks to condition an INR-based GAN model on text. In this setting, the generator and the discriminator weights are controlled by their corresponding HyperNetworks, which modulate weight parameters using the provided text query. We propose an effective Word-level hyper-modulation Attention operator, termed WhAtt, which encourages grounding words to independent pixels at input $(x, y)$ coordinates. To the best of our knowledge, our work is the first that explores Text to Continuous Image Generation (T2CI). We conduct comprehensive experiments on COCO $256^2$, CUB $256^2$, and ArtEmis $256^2$ benchmark, which we introduce in this paper. HyperCGAN improves the performance of text-controllable image generators over the baselines while significantly reducing the gap between text-to-continuous and text-to-discrete image synthesis. Additionally, we show that HyperCGAN, when conditioned on text, retains the desired properties of continuous generative models (e.g., extrapolation outside of image boundaries, accelerated inference of low-resolution images, out-of-the-box super-resolution). Code and ArtEmis $256^2$ benchmark will be made publicly available.

## 1 INTRODUCTION

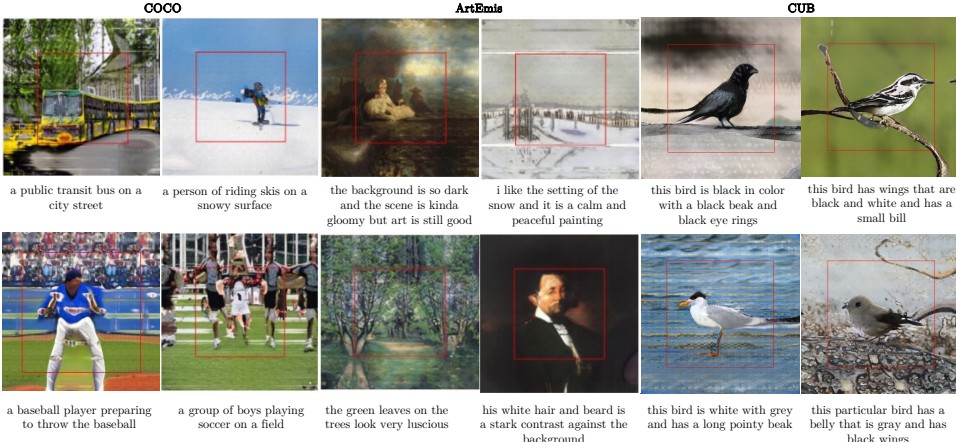

Figure 1: **Text Conditioned Extrapolation outside of Image Boundaries**: The red rectangles indicate the resolution boundaries that our HyperCGAN model was trained. On three datasets, our model can synthesize meaningful pixels at surrounding $(x, y)$ coordinates beyond these boundaries.

Humans have the innate ability to connect what they visualize with language or textual descriptions. Text-to-image (T2I) synthesis, an AI task inspired by this ability, aims to generate an image conditioned on a textual input description. Compared to other possible inputs in the conditional generation literature, sentences are an intuitive and flexible way to express visual content that we may want to generate. The main challenge in traditional T2I synthesis lies in learning from the unstructured description and connecting the different statistical properties of vision and language inputs. This field has seen significant progress in recent years in synthesis quality, the size and complexity of datasets used as well as image-text alignment (Xu et al., 2018; Li et al., 2019; Zhu et al., 2019; Tao et al., 2022; Zhang et al., 2021; Ramesh et al., 2021).

Existing methods for T2I can be broadly categorized based on the architecture innovations developed to condition on text. Models that condition on a single caption input include stacked architectures (Zhang et al., 2017), attention mechanisms (Xu et al., 2018), Siamese architectures (Yin et al., 2019), cycle consistency approaches (Qiao et al., 2019), and dynamic memory networks (Zhu et al., 2019). A parallel line of work (Yuan & Peng, 2019; Souza et al., 2020; Wang et al., 2020) looks at adapting unconditional models for T2I synthesis. Despite the significant progress, images in existing approaches are typically represented as a discrete 2D pixel array which is a cropped, quantized version of the true continuous underlying 2D signal. We take an alternative view, in which we use an implicit neural representation (INR) to approximate the continuous signal. This paradigm accepts coordinate locations $(x, y)$ as input and produces RGB values at the corresponding location for the continuous images. Working directly with continuous images enables several useful features such as extrapolation outside of image boundaries, accelerated inference of low-resolution images and out-of-the-box superresolution. Our proposed network, the HyperCGAN uses a HyperNetwork-based conditioning mechanism that we developed for Text to continuous image generation. It extends the INR-GAN (Skorokhodov et al., 2021a) backbone to efficiently generate continuous images conditioned on input text while preserving the desired properties of the continuous signal. Figure 1 shows examples of images generated by our HyperCGAN model on the CUB (Wah et al., 2011), COCO(Lin et al., 2015), and ArtEmis (Achlioptas et al., 2021) datasets. By design and while conditioning on the input text, we can see that HyperCGAN, trained on the CUB dataset, can extend bird images with more natural details like the tail and background (see Figure 1) and the branches (top right). We observe similar behavior on scene-level benchmarks, including COCO and ArtEmis (introduced in this paper).

By representing signals as continuous functions, INRs do not depend on spatial resolution. Thus, the memory requirements to parameterize the signal grow not with respect to spatial resolution but only increase with the complexity of the signal. This type of representation enables generated images to have arbitrary spatial resolutions, keeping the memory requirements near constant. In contrast, discrete-based models need both generator and discriminator to scale with respect to spatial resolution, making training of these models impractical. Figure 2 shows that for discrete-based models, increasing training resolution leads to decreasing effective batch size during training due to GPU memory limits. Coupled with the expressiveness of HyperNetworks, we believe that building conditional generative models that are naturally capable of producing images of arbitrary resolutions while maintaining visual semantic consistency at low training costs is a promising paradigm in the fu-

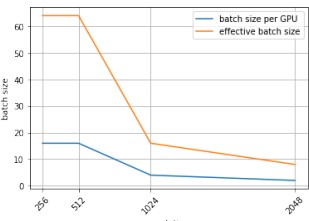

Figure 2: Scalability limitations in discrete decoders: Increasing training resolution decreases batch size/GPU hitting GPU limits.

ture progress of generative models. Our work introduces a step in this direction. The prevalent T2I models in the literature like AttnGAN (Xu et al., 2018), ControlGAN (Li et al., 2019) and XMC-GAN (Zhang et al., 2021) use architecture-specific ways to condition the generator and discriminator on textual information and often introduce additional text-matching losses. These approaches use text embeddings $c$ to condition their model by updating a hidden representation $h$. Unlike these approaches, we explore a different paradigm in HyperCGAN, and use HyperNetworks (Ha et al., 2016) to condition the model on textual information $c$ by modulating the model weights. Such a procedure can be viewed as creating a different instance of the model for each conditioning vector $c$ and was recently shown to be significantly more than the embedding-based conditioning approaches. (Galanti & Wolf, 2020). A traditional HyperNetwork (Chang et al., 2020) generates the entire parameter vector $\theta$ from the conditioning signal $c$ ie. $\theta = F(c)$, where $F(c)$ is a *modulating* HyperNetwork. However, this quickly becomes infeasible in modern neural networks where $|\theta|$ can easily span millions of parameters. Our HyperNetwork instead produces a *tensor-decomposed* modulation

$F(\boldsymbol{c}) = \boldsymbol{M}$ of the same size as the weight tensor $\boldsymbol{W}$. This tensor is then used to alter $\boldsymbol{W}$ via an elementwise multiplicative operation $\boldsymbol{W}_c = \boldsymbol{W} \odot \boldsymbol{F}(\boldsymbol{c})$. Additionally, we develop an attention-based word level modulation WhAtt to alter weight tensors $\boldsymbol{W}$ of both Generator and Discriminator using $F(\boldsymbol{c})$. Our primary contributions are as follows:

- We propose the HyperCGAN framework for synthesizing continuous images from text input. The model is augmented with a novel language-guided mechanism termed *WhAtt*, that modulates weights at the word level.

- We show that our method has the ability to meaningfully extrapolate outside the image boundaries, and can outperform most existing discrete methods on the COCO and ArtEmis datasets, including stacked generators and single generator methods.

- We establish a baseline on a new affective T2I benchmark based on the ArtEmis dataset (Achlioptas et al., 2021), which has 455,000 affective utterances collected on more than 80K artworks. ArtEmis contains captions that explain emotions elicited by a visual stimulus, which can lead to more human emotion-aware T2I synthesis generation models.

## 2 RELATED WORK

**Text-to-Image Generation.** T2I synthesis has been an active area of research since at least (Mansimov et al., 2015; Reed et al., 2016a) proposed a DRAW-based (Gregor et al., 2015) model to generate images from captions. (Reed et al., 2016a) first demonstrated improved fidelity of the generated images from text using GANs (Goodfellow et al., 2014). Several GAN-based approaches for T2I synthesis have emerged since. StackGAN (Zhang et al., 2017) proposed decomposing the T2I generation into two stages - a coarse to fine approach and used conditional augmentation of the conditioning text. Later, (Xu et al., 2018) proposed AttnGAN, an extended version of StackGAN, and adopted cross-modal attention mechanisms for improved visual-semantic alignment and grounding. Following the architecture of AttnGAN, some approaches were proposed to improve the generation quality (Li et al., 2019; Zhu et al., 2019). XMC-GAN (Zhang et al., 2021), DF-GAN (Tao et al., 2022) proposes to use additional auxiliary losses to improve visual semantic alignment. Non-GAN based generative models have also been explored in T2I, e.g autoregressive approaches (Reed et al., 2016b; 2017; Ramesh et al., 2021; Gafni et al., 2022), flow-based models (Mahajan et al., 2020).

**Diffusion Models.** With the introduction of diffusion models (Sohl-Dickstein et al., 2015; Song & Ermon, 2019; Ho et al., 2020), which learns to perform denoising task, the breakthrough has been made in T2I due to the emergence of diffusion-based models conditioned on text (Ramesh et al., 2022; Nichol et al., 2021; Saharia et al., 2022; Rombach et al., 2022; Gu et al., 2022). These methods cannot be directly compared with our work since they have a huge number of parameters and requires a massive amount of data for training. The diffusion-based methods do not suffer from mode collapse, but their compute cost and carbon footprint are much higher than GAN-based approaches.

**Art generation.** Synthetically generating realistic artwork with conditional GAN is challenging due to unstructured shapes and its metaphoric nature. Several works have explored learning artistic style representations. ArtGAN (Tan et al., 2017; 2018) trained a conditional GAN on artist, genre, and style labels. (Alvarez-Melis & Amores, 2017) proposed emotion-to-art generation by training an AC-GAN (Odena et al., 2017) on ten classes of emotions. Another line of work includes CAN (Elgammal et al., 2017) and later H-CAN (Sbai et al., 2018), which generates creative art by learning about styles and deviating from style norms. We extend prior work by applying our HyperNetwork-based conditioning to the novel text-to-continuous-image generation task on the challenging ArtEmis (Achlioptas et al., 2021) dataset, where we leverage verbal explanations as conditioning signals to achieve better human cognition-aware T2I synthesis.

**Implicit Neural Representation (INR).** INRs parametrize any type of signal (e.g. images, audio signals, 3D shapes) as a continuous function that maps the domain of the signal to values at a specified coordinate (Genova et al., 2019; Mildenhall et al., 2020; Sitzmann et al., 2019; 2020). For 2D image synthesis, several works have explored ways to enable INRs using generative models (Anokhin et al., 2021; Skorokhodov et al., 2021a;b).

**Connection to HyperNetworks.** HyperNetworks are models that generate parameters for other models. They have been applied to several tasks in architecture search (Zhang et al., 2019), few-

shot learning (Bertinetto et al., 2016), and continual learning (von Oswald et al., 2020). Generative HyperNetworks, also called implicit generators(Skorokhodov et al., 2021a; Anokhin et al., 2021) were recently shown to rival StyleGAN2 (Karras et al., 2020) in generation quality. Our HyperC-GAN generates continuous images conditioned on text using two types of Hypernetworks: (1) Image generator HyperNetwork, which produces an image represented by its INR. (2) Text controlling HyperNetwork that guides the learning mechanism of the image generator HyperNetwork using the input text. Despite the progress in unconditional INR-based decoders (e.g., (Lin et al., 2019; Skorokhodov et al., 2021a; Anokhin et al., 2021; Skorokhodov et al., 2021b)), generating high-quality continuous images conditioned on text is less studied compared to discrete image generators. Our HyperNetwork-augmented modulation approach facilitates conditioning the continuous image generator on text while preserving the desired INR properties (e.g., out-of-the-box-super resolution, extrapolation outside image boundaries).

## 3 Approach

The T2I generation task can be formulated as modeling the data distribution of images $\mathbb{P}_r$ given a conditioning signal $c$. We use a standard GAN training setup where we model the image distribution using a generator $G$. In our case, $c$ is text information in the form of sentence, or word embeddings. During training, we alternate between optimizing the generator and discriminator objectives:

$$
\begin{aligned}
L_{\mathrm{D}}(c) &= -\mathbb{E}_{x \sim \mathbb{P}_r}[D(x, c)] - \mathbb{E}_{G(z,c) \sim \mathbb{P}_g}[1 - D(G(z, c), c)] \\
L_{\mathrm{G}}(c) &= -\mathbb{E}_{G(z,c) \sim \mathbb{P}_g}[D(G(z, c), c)] + \lambda L_{\mathrm{contrastive}}
\end{aligned}
\tag{1}
$$

where $\mathbb{P}_g$ is the generated image distribution, and $\mathbb{P}_r$ is its real distribution. $L_{\mathrm{D}}(c)$ and $L_{\mathrm{G}}(c)$ are the discriminator and the generator losses, respectively. To facilitate continuous image generation, HyperCGAN augments the unconditional baseline INR-GAN(Skorokhodov et al., 2021a) with an effective modulation mechanism that encourages better sentence-level and word-level alignment. To encourage fine-grained image-text matching, the generator is regularized with an auxiliary contrastive loss based on the Deep Attentional Multimodal Similarity Model (DAMSM) (Xu et al., 2018), which measures the similarity between generated images and global sentence-level as well as fine-grained word-level information. As shown in our experiments, our proposed modulation helps DAMSM loss improve continuous image-text alignment at the word level while preserving high image fidelity. We also explore integrating the CLIP (Radford et al., 2021) loss to improve the alignment between the text and the generated continuous images. The following subsections introduce INR-based decoders and describe how we adapted them in our HyperCGAN approach to facilitate continuous image generation conditioned on text.

### 3.1 INR-based Generator Backbone: INR-GAN (Skorokhodov et al., 2021a; Anokhin et al., 2020)

INR is an implicit approach that can represent a 2D image, with a neural network to produce RGB pixel values given image coordinate locations $(x, y)$. We build our approach upon the INR-based generator (Skorokhodov et al., 2021a; Anokhin et al., 2020), that consists of two main modules: a hypernetwork $H(z)$ and an MLP model $F_{\theta(z)}(x, y)$. The hypernetwork $H(z)$ samples a noise vector $z \sim \mathcal{N}(0, I)$ and generates parameters for an MLP model $F_{\theta(z)}(x, y)$. The MLP model $F_{\theta(z)}(x, y)$ then predicts RGB values at each location $(x, y)$ of a predefined coordinate grid to synthesize an image. The weights of the MLP model are modulated through a Factorized Multiplicative Modulation (FMM) mechanism, where two matrices are multiplied together and passed through an activation function to obtain a modulating tensor. Later, this modulating tensor is multiplied by the shared parameters matrix of the MLP network.

### 3.2 Hyper-Conditional GANs (HyperCGANs)

**Architecture Overview.** Our generator architecture is based on the multi-scale INR-GAN and mainly consists of fully-connected linear layers followed by activations. The weights of these layers are two-dimensional. i.e., $W^\ell \in \mathbb{R}^{c_{\mathrm{out}} \times c_{\mathrm{in}} \times 1 \times 1}$ at layer $l$. We use the StyleGAN2 discriminator during the training process, which comprises a series of ConvNet blocks. The convolutional weights can be represented as a four-dimensional tensor $W^\ell \in \mathbb{R}^{c_{\mathrm{out}} \times c_{\mathrm{in}} \times k_h \times k_w}$. In the INR-GAN, these

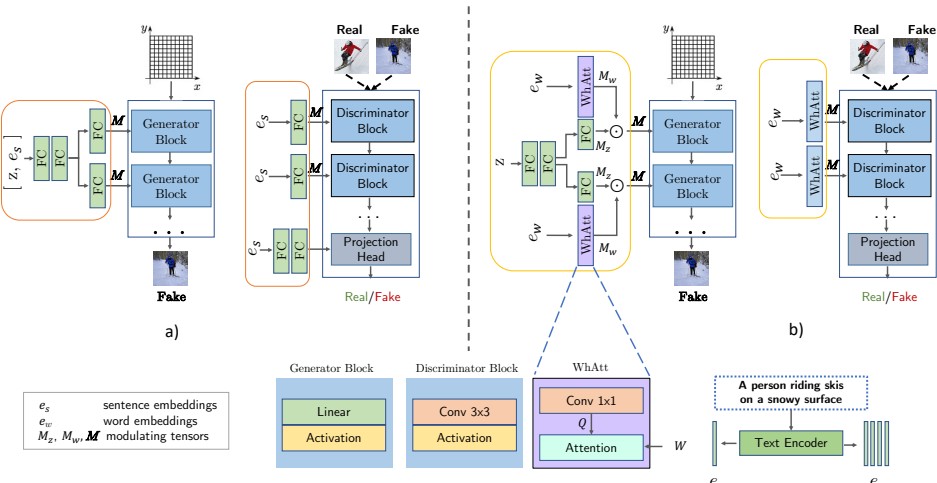

Figure 3: **The architecture of the proposed HyperCGAN,** with two ways of conditioning. **a) Sentence-Level Modulation:** Generator is conditioned with a hypernetwork which takes concatenation of noise vector $z$ and sentence embedding $e_s$. Then, the weights of every Linear layer of the generator are modulated by modulating tensor. Discriminator's convolutional weights at block $l$ are modulated by the hypernetwork operating at level $l$. The final projection head is conditioned as $h^T F(e_s)$, where $F(e_s)$ is two-layer MLP and h is the output of the last discriminator block. **b) Word-Level Modulation with WhAtt attention:** Two hypernetworks are used to condition the generator. The first is MLP which receives noise vector $z$ and outputs modulating tensor $M_z$, and the second is Conv1x1 followed by a Word-level hyper-Attention mechanism proposed in this work, dubbed as WhAtt. Details are introduced in Section 3.2.2.

weights are not conditioned on the text. As our *goal* is to condition both generator and discriminator on input text $c$, we apply the HyperCGAN conditional modulation framework to both the INR-GAN generator and discriminator. In this framework, $c$ is transformed by a HyperNetwork to produce modulating tensors for the weight tensors. Figure 3 is an overview of our proposed HyperCGAN approach.

### 3.2.1 Leveraging conditional signal for weight modulation

When conditioning the generator, we use two strategies to generate the modulating tensor $M$ for linear layers depending on the language representation granularity (word-level or sentence level). **Sentence-level Conditioning:** We also explore sentence-level conditioning on top of sentence embeddings $e_s$. In this case, the HyperNet backbone receives as input the concatenation of noise vector $z \sim \mathcal{N}(0, I)$ of size $d_z$ and sentence embedding vectors $e_s$ of size $d_c$; i.e., $[z, s]$. Then, for each linear layer $\ell$ in the INR MLP-decoder, separate modulating tensors $M_{z,s}^l$ are generated through fully-connected layers (FC) (see Figure 3.a). This tensor $M_{z,s}^l$ is further used to modulate the generator's weight $W_G^\ell$ at layer $\ell$ through element-wise multiplication; see Equation 5.

**Word-level Conditioning:** Word embeddings $e_w \in \mathbb{R}^{\Omega \times d_w}$ are represented as a sequence of individual vectors of size $d_w$ for each word in the sentence, where $\Omega$ denotes sequence length of the word embeddings (i.e., the number of words). Two hypernetworks are used to condition the generator. The first is an MLP which receives noise vector $z$ and outputs the modulating tensor $M_z$, and the second is a Conv1x1 followed by a novel Word-level Hyper-Attention mechanism proposed in this work, termed WhAtt, detailed later in this section.

Slightly different from the generator, hypernetworks of the discriminator are either FC which takes sentence embedding $e_s$ as an input and generate a tensor $M_s^l$ or Conv1x1 which receives word embedding $e_w$ and generate a tensor $M_w^l$ for modulation.(see Figure 3.a,b). In addition, the final projection head in the discriminator is conditioned through $s = h^\top F(e_s)$, where $h$ is the output of the last discriminator block and $F(e_s)$ is the vector produced by our hypernetwork. This form resembles the traditional Projection Discriminator (Miyato & Koyama, 2018) that uses output $s =$

$h^\top j$, (j one-hot), which we generalize to condition on beyond one-hot class labels (see dsicriminator in Figure 3.a).

**Extreme Modulating Tensor Factorization.** Producing a full-rank tensor $M^\ell$ for each block $l$ is memory-intensive and infeasible even for modestly sized architectures. For example, if the hidden layer size of our hypernetwork is of size $d_h = 512$ and the convolutional weight tensor at layer $\ell$ is of dimensionality $d_o = c_{\text{out}} \times c_{\text{in}} \times k_h \times k_w = 512 \times 512 \times 3 \times 3 \approx 2.4$ million, then the output weight matrix in the hypernetwork will be of size $d_o \times d_h = 1.2$ billion. To overcome this issue, we propose factorizing the modulating tensor with an *extreme* low-rank tensor decomposition for learning efficiency. The canonical polyadic (CP) decomposition (Kiers, 2000) lets us express a rank-$R$ tensor $\mathcal{T} \in \mathbb{R}^{d_1 \times \dots \times d_n}$ as a sum of $R$ rank-1 tensors:

$$\mathcal{T} = \sum_{r=1}^{R} \mathbf{t}_1^r \otimes \dots \otimes \mathbf{t}_n^r \tag{2}$$

where $\otimes$ is the tensor product and $\mathbf{t}_r^k$ is a vector of length $d_k$. Going back to our example mentioned above, if we instead generate separately low-rank factors and build modulating tensor out of the factors $d_o = c_{\text{out}} + c_{\text{in}} + k_h + k_w = 512 + 512 + 3 + 3 = 1030$. So, the output weight matrix in the hypernetwork will be of size $d_o \times d_h = 527360$ which leads to $\approx 99.95\%$ decrease in the parameter size of hypernetworks. Therefore, $M_{z,s}^l$ will be the tensor product of 4 low-rank rank-1 tensors $\mathbf{t}_1$, $\mathbf{t}_2$, $\mathbf{t}_3$, and $\mathbf{t}_4$ of size $c_{\text{out}}$, $c_{\text{in}}$, $k_h$ and $k_w$, respectively.

### 3.2.2 Word-level Modulation with WhAtt Attention

In contrast to sentence embedding where words are summarized in one vector, individual word embeddings consist of sequences of individual word encodings, containing fine-grained information that is typically visually grounded to the image. Hence, we focus on how to leverage this information in our model. We introduce a Word-level Hyper Attention mechanism, denoted as WhAtt, that can leverage this word-level as well as sentence-level information through self-attention.

*WhAtt Attention.* First, word embeddings from the text encoder are extracted. These embeddings are of size $\Omega \times d$ where $\Omega$ denotes sequence length of the word embeddings (i.e., the number of words) and $d$ is an embedding size. The word embeddings are further encoded with a different HyperNetwork which consists of a single `Conv1x1` layer for each layer $l$. From every hypernetwork at layer $l$, a different tensor $\mathcal{T}^\ell \in \mathbb{R}^{\Omega \times (c_{\text{in}} + k_h + k_w)}$ is obtained. Basically, a tensor $\mathcal{T}^\ell$ is composed of $\Omega$ number of different vectors $\mathbf{v}^i \in \mathbb{R}^{c_{\text{in}} + k_h + k_w}$ corresponding to the $i$-th word. Then, each vector $\mathbf{v}^i$ is "sliced" into three low-rank factors $\mathbf{v}_{\text{in}}^i$, $\mathbf{v}_h^i$, $\mathbf{v}_w^i$ of dimensions $c_{\text{in}}$, $k_h$, $k_w$, respectively. From the entire tensor $\mathcal{T}^\ell$, we, therefore, derive a two dimensional matrix $Q^\ell \in \mathbb{R}^{\Omega \times (c_{\text{in}} \times k_h \times k_w)}$ using tensor factorization (Eq. 2) which can be expressed via an outer product operation:

$$Q_i^\ell = \mathbf{v}_{\text{in}}^i \otimes \mathbf{v}_h^i \otimes \mathbf{v}_w^i \tag{3}$$

where $Q_i^\ell$ is the $i$-th row in $Q^\ell$, We apply scaled dot product attention mechanism (Vaswani et al., 2017) to attend to the relevant words in the resulting tensor $M_w^\ell \in \mathbb{R}^{\Omega \times c_{\text{in}} \times k_h \times k_w}$:

$$M_w^\ell = \text{WhAtt}(W^\ell, Q^\ell) = \text{softmax}\left(\frac{W^\ell (Q^\ell)^T}{\sqrt{c_{\text{out}}}}\right) Q^\ell, \tag{4}$$

where $W^\ell$ is the weight matrix at layer $l$, $M_w^\ell$ is the word-level modulating tensor, $W^\ell$ and $M_w^\ell \in \mathbb{R}^{c_{\text{out}} \times c_{\text{in}} \times k_h \times k_w}$. Finally, the modulating tensors for the generator and discriminator for both sentence and word based modulation are defined by Eq. 5 and Eq. 6, respectively:

$$\begin{aligned} \hat{W}_G^\ell &= M_{z,s}^\ell \odot W_G^\ell \\ \hat{W}_D^\ell &= M_s^l \odot W_D^\ell \end{aligned} \tag{5} \qquad\qquad \begin{aligned} \hat{W}_G^\ell &= M_z^\ell \odot M_w^\ell \odot W_G^\ell \\ \hat{W}_D^\ell &= M_w^l \odot W_D^\ell \end{aligned} \tag{6}$$

where $\hat{W}_G^\ell$ and $\hat{W}_D^\ell$ are the modulated weights at layer $\ell$ for the generator and the discriminator respectively. $W_G^\ell$ and $W_D^\ell$ are the corresponding weights at layer $\ell$ before modulation. Note that like sentence level modulation, $k_h = 1$ and $k_w = 1$ for the generator and are equal to the kernel size in the discriminator as it is convolutional.

Our word-level modulation aims at grounding words to independent pixels at input $(x, y)$ coordinates, represented as low-res features in the earlier layers, and the final RGB value in the last

layer. More generally, word-level conditioning benefit for visual-semantic consistency was first demonstrated for discrete decoders in AttnGAN (Xu et al., 2018). Our word-level modulation is our proposed mechanism to bring similar properties to text-conditioned continuous image generation.

## 4 EXPERIMENTS AND RESULTS

In this section, we first define the used datasets, metrics, and our baselines following which we compare our model relative to the baselines on the benchmarks, and study the various properties and limitations of our approach.

**Datasets.** We comprehensively evaluate HyperCGAN on the challenging MS-COCO (Lin et al., 2015), ArtEmis (Achlioptas et al., 2021), and CUB (Wah et al., 2011) datasets.

– **COCO** $256^2$ contains over 80K images for training and more than 40K images for testing. Each image has 5 associated captions that describe the visual content of the image. We use the splits proposed in (Xu et al., 2018) to train and test our models.

– **ArtEmis** $256^2$ **(introduced T2I benchmark)** contains over 450K emotion attributes and explanations from humans on more than 81K artworks from WikiArt dataset. Each image is associated with at least 5 captions. The unique aspect of the dataset is that utterances are more affective and subjective rather than descriptive. These aspects of the dataset impose additional challenges on our T2I generation task. We use the train and test splits provided by the authors and benchmark recent T2I methods on it. Both COCO and ArtEmis are scene-level T2I benchmarks.

– **CUB** $256^2$ contains 8,855 training and 2,933 test images of bird species. Each image has 10 corresponding text descriptions. In contrast to COCO and ArtEmis, CUB is an object-level benchmark, yet challenging as the bird species are fine-grained.

**Evaluation Metrics**. We evaluate all models in terms of both Image Quality and Text-Image Alignment. Due to the limitations of the Inception score (IS) (Salimans et al., 2016) to capture the diversity and quality of the generation, we report Frechet Inception Distance (FID) (Heusel et al., 2017) score following previous works (Zhang & Schomaker, 2020; Tao et al., 2022). Additionally, we compute *R-precision* since image quality scores alone cannot reflect whether the generated image is well conditioned on the given text description. Given a generated image, R-precision measures the retrieval rate for the corresponding caption using a surrogate multi-modal network which computes the similarity score between image features and text features. (Zhu et al., 2019; Xu et al., 2018; Li et al., 2019; Zhu et al., 2019) relied on a pretrained DAMSM model consisting of a text encoder and image encoder to compute the similarity between generated image and text descriptions for R-precision, termed as DAMSM-R. However, the same DAMSM model used during training and evaluation leads to severely biased behavior towards this metric (see Table 9 in Appendix). Therefore, as suggested in (Park et al., 2021), we also report R-precision score where image-text similarity is computed with CLIP (Radford et al., 2021), dubbed as CLIP-R. Moreover, we conduct a human evaluation to assess the meaningfulness and image-text alignment quality for the extrapolated regions facilitated by the conditional continuous image generation ability of HyperCGAN.

**Text to Continuous Image (T2CI) Generation Baselines.** Since our work is the first attempt on T2CI, we define the following baselines: **INR-CGAN**$^{sent}$: we transform unconditional INR-GAN to be conditioned on sentence embeddings as a baseline. In this transformation, this baseline simply takes the concatenated noise vector and sentence embeddings and then generates parameters for the decoder to synthesize an image. We condition its discriminator via a projection head like in our approach but do not modulate the convolution layers conditioned on text. This corresponds to configuration B in Table 1. **HyperCGAN**$^{sent}$: This baseline is built on top of INR-CGAN$^{sent}$. The generator stays unchanged, but the discriminator convolution weights are modulated with our "Efficient Sentence level Modulation" (config E in Table 1). **HyperCGAN**$^{word}$: The generator and discriminator of this model are conditioned via our proposed WhAtt mechanism (config H in Table 1). For both HyperCGAN$^{sent}$ and HyperCGAN$^{word}$, we either use DAMSM-based or CLIP-based regularizers. When the model is trained with one of these regularizers, we indicate it with subscript, e.g., HyperCGAN$^{word}_{DAMSM}$ or HyperCGAN$^{word}_{CLIP}$.

**T2CI Results.** In Table 1, INR-CGAN$^{sent}$ (config B) with naive conditioning achieves for 34.91% for CLIP-R and 27.73 in terms of FID. When its discriminator changed to hyper-modulated one,

| Configuration | | name ↑ | CLIP-R ↑ | FID ↓ |
|---|---|---|---|---|
| A | Unconditional INR-GAN (Skorokhodov et al., 2021a) | INR-GAN | NA | 24.74 |
| B | + sentence conditioning | INR-CGAN$^{sent}$ | 34.91% | 27.73 |
| C |    + DAMSM regularizer | INR-CGAN$^{sent}_{DAMSM}$ | 57.11% | 18.51 |
| D |    + CLIP regularizer | INR-CGAN$^{sent}_{CLIP}$ | 49.71% | 21.88 |
| E | + hyper-modulated D | HyperCGAN$^{sent}$ | 40.81% | 28.29 |
| F |    + DAMSM regularizer | HyperCGAN$^{sent}_{DAMSM}$ | 51.34% | 29.66 |
| G |    + CLIP regularizer | HyperCGAN$^{sent}_{CLIP}$ | 55.41% | 28.83 |
| H | + our word-conditioned WhAtt attention | HyperCGAN$^{word}$ | 37.23% | 25.39 |
| L |    + DAMSM regularizer | HyperCGAN$^{word}_{DAMSM}$ | 64.14% | 18.58 |
| M |    + CLIP regularizer | HyperCGAN$^{word}_{CLIP}$ | 63.99% | 27.21 |

Table 1: **T2CI Performance on COCO** $256^2$. Our hypernetwork-based conditioning makes it possible to use word-level conditioning, which is crucial in achieving good results. Blue is for the best result and green for 2nd best. Note that CLIP-R is meaningless for unconditional INR-GAN (config A). Note that config F starts from unconditional INR-GAN, similar to config B and C.

CLIP-R is improved to 40.81%, and show comparable FID score 28.29 (config E). When both INR-decoder and discriminator is conditioned via our WhAtt method on word embeddings (config H), FID score improves from 28.25 to 25.39 and achieves slightly lower CLIP-R score 37.23% compared to config B. Our WhAtt conditioning coupled with contrastive regularizers achieves CLIP-R retrieval scores are improved much leading to the best scores 64.14% and 63.99%, while achieving the better FID scores 18.91 and 27.21 for HyperCGAN$^{word}_{DAMSM}$ and HyperCGAN$^{word}_{CLIP}$, respectively configs L and M. The word-level modulation has significantly better performance due to the improved granularity connecting the generated images to the input text. It is interesting to observe that HyperCGAN$^{word}_{DAMSM}$ even outperforms Unconditional INR-GAN by 5.83 FID points. Despite that regularizers focus more on visual semantic alignment (CLIP-R) than image quality (FID), we observe relative improvement also on FID in most cases, which could be due to the improved representation guided by text.

**WhAtt Attention Generalization on Discrete Decoders.** Our sentence-based modulation and word-level WhAtt conditioning mechanism can easily be applied to conventional convolution-based generators. In this case, our hypernetwork-based methods modulate the convolution weights of the generator, which has convolutional layers with kernel sizes more than 1 ($k_h > 1$ and $k_w > 1$). To show this, we enable the standard unconditional Style-GAN2 (Karras et al., 2020) backbone to be conditioned on

| Model | CLIP-R ↑ | FID ↓ |
|---|---|---|
| HyperC-SG$^{sent}_{DAMSM}$ | 54.45% | 31.47 |
| HyperC-SG$^{word}_{DAMSM}$ | **61.49%** | **20.81** |
| HyperCGAN$^{sent}_{DAMSM}$ | 51.34% | 29.66 |
| HyperCGAN$^{word}_{DAMSM}$ | **64.14%** | **18.58** |

Table 2: Discrete and continuous synthesis performance with our hypernet-based conditioning on COCO $256^2$.

either sentence or word embeddings dubbed as Table 2 shows that HyperC-SG$^{word}_{DAMSM}$ equipped with our proposed WhAtt mechanism boosts CLIP-R results to 61.49% compared to HyperC-SG$^{sent}_{DAMSM}$ at 54.45%, while also significantly improving the image quality with FID score 20.81 from 31.47. We also include the same comparison in Table 2 our continuous model HyperCGAN, which achieves the best results, and the WhAtt mechanism improvement is more significant (18.58 FID, 64.14 CLIP-R).

**Properties of our conditional INR-based decoder**. **a)** *Interpolation*. We train our model in downsampled $128^2$ images and conduct either INR superresolution with a denser coordinate grid or standard upsampling techniques for comparison. Experimental results indicate our model, beating classical interpolation methods on all datasets by 2.35 points on average (see Table 8 and Figure 9 for results). **b)** *Extrapolation*. Figure 1 shows the ability of HyperCGAN to extrapolate outside of the training image boundaries. After training on a coordinate grid with a specific range, HyperCGAN can be evaluated on a wider grid. We are interested in studying whether the extended

| Dataset | meaningful | not sure | not meaningful |
|---|---|---|---|
| COCO | 68.8% | 10.4% | 20.8% |
| ArtEmis | 75.6% | 8% | 16.4% |

Table 3: Human Subject Experiment on Extrapolation meaningfulness.

| Dataset | more aligned or same | less aligned |
|---|---|---|
| COCO | 66.4% | 33.6% |
| ArtEmis | 69.2% | 30.8% |

Table 4: Human Subject Experiment on Extrapolation alignment with text.

regions made sense beyond the training data coordinates. To validate this, we conducted user studies where subjects were asked to indicate a) whether the extended area in the generations is meaningful and b) whether the extended area makes the image more aligned with the text description. In Table 3, for COCO, the results show that 68.8% of responses indicate that the out-of-the-region extrapolation is meaningful while 20.8% of them say it is not. For ArtEmis, 75.6% of responses are in favor of

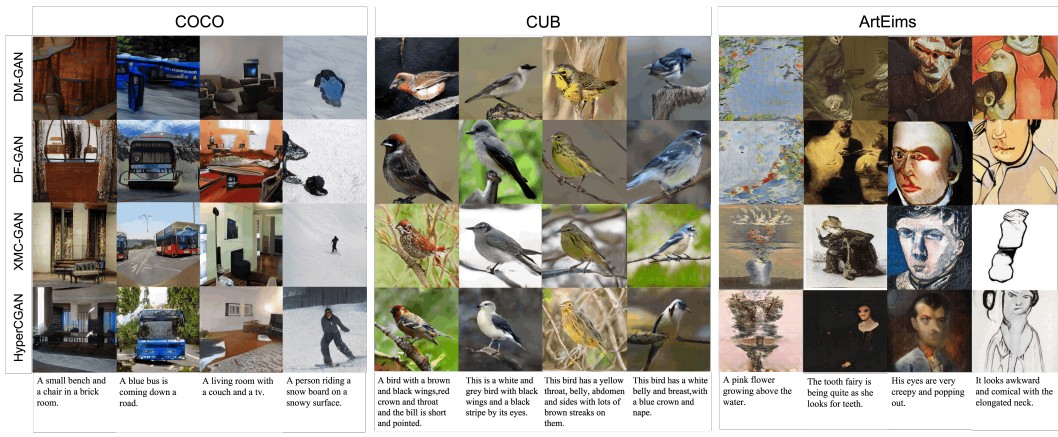

Figure 4: HyperCGAN qualitative results on COCO $256^2$, CUB $256^2$, and ArtEmis $256^2$.

meaningful; meanwhile, 16.4% of them show the opposite. In Table 4, more than 65% of responses suggested that the alignment between the image and text description improved or remained the same for both COCO and ArtEmis.

**Comparison to the State-of-the-Art.** To demonstrate the gap we reduced compared to T2I discrete decoder, We compare HyperCGAN with discrete state-of-the-art approaches (Xu et al., 2018; Li et al., 2019; Zhu et al., 2019; Zhang et al., 2021; Tao et al., 2022). Note that AttnGAN(Xu et al., 2018) and DM-GAN (Zhu et al., 2019) are multi-stage generations. Figure 4 shows qualitative results of our model compared to baselines. Generation qualities are comparable to the state-of-the-art. Table 5 shows that our models achieve the highest CLIP-R on COCO and comparable results to XMC-GAN on ArtEmis and CUB. For fair comparison to other baselines that use DAMSM regularizer, we report the scores with our HyperCGAN$_{DAMSM}^{word}$ model, which achieves higher CLIP-R 64.14% in COCO and 16.26%. Note that every baseline except DF-GAN utilizes both sentence and word embeddings, while our model is only conditioned on one type of text embeddings during training and still achieves superior results compared to other baselines in terms of FID on all datasets (except XMC-GAN, 2.5 times more model parameters than ours). As for CUB dataset, HyperCGAN $_{DAMSM}^{word}$ with WhAtt achieves the best FID score of 11.00. Compared to XMC-GAN on Artemis and CUB, results in terms of FID and CLIP-R are almost the same. However, our model requires much fewer number of parameters, making it more efficient during training. Almost all the baselines'

results exceed CLIP-R score of real images, and we argue that CLIP model might be good at scene-level recognition rather than fine-grained object-level recognition. We also observe that our model outperforms the large-scale T2I model DALL-E (Ramesh et al., 2021) on COCO and CUB: 27.21 vs. 27.50 and 11.00 vs. 56.10, respectively. However, the comparison to DALL-E might not be fair since their result is based on zero-shot T2I generation. DALL-E1 was trained on a large amount of data, orders of magnitudes larger than the benchmarks we are using, and it may/may not cover data similar to these benchmarks

| Model | COCO $256^2$ | | ArtEmis $256^2$ | | CUB $256^2$ | | |
|---|---|---|---|---|---|---|---|
| | FID ↓ | CLIP-R ↑ | FID ↓ | CLIP-R ↑ | FID ↓ | CLIP-R ↑ | NoP ↓ |
| AttnGAN (Xu et al., 2018) | 35.49 | 29.31% | 45.64 | 7.11% | 23.98 | 31.23% | 230M |
| ControlGAN (Li et al., 2019) | 34.52 | 24.96% | 42.01 | 7.38% | 22.85 | 35.71% | 250M |
| DM-GAN (Zhu et al., 2019) | 32.64 | 40.31% | 31.4 | 12.92% | 16.09 | **45.07%** | 46M |
| XMC-GAN (Zhang et al., 2021) | **9.87** | 48.31% | 15.47 | **36.68%** | 15.56 | 30.40% | 166M |
| DF-GAN (Tao et al., 2022) | 19.32 | 26.13% | 25.4 | 9.81% | 14.81 | 28.39% | 19M |
| HyperCGAN$_{DAMSM}^{word}$ (continuous) | 18.58 | **64.14%** | 19.83 | 16.26% | **11.00** | 30.51% | 65M |
| HyperCGAN$_{CLIP}^{word}$ (continuous) | 27.21 | 63.99% | **15.40** | 34.63% | 16.48 | 19.02% | 65M |
| Real Images | - | 89.43% | - | 45.12% | - | 26.20% | |

Table 5: Comparison to SOTA Discrete T2I models.

## 5 CONCLUSION

In this paper, we propose HyperCGAN, a novel HyperNet-based conditional continous GAN. Hyper-CGAN is a text-to-continuous-image generative model with a single generator that operates with a novel language-guided tensor modulation operator for sentence-level and word-level attention mechanism. To our knowledge, HyperCGAN is the first approach that facilitates text-to-continuous-image generation, and we show its ability to meaningfully extrapolate images beyond training image di-

mension while maintaining the alignment with the input language description. We showed that HyperCGAN achieves better performance compared to existing discrete-based text-to-image synthesis baselines. In addition, we demonstrated that our hypernet-modulation methods can be applied to discrete GANs as well. We hope that our method may encourage future work on hyper networks on Text to Continuous Image Generation (T2CI).

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

# 6 APPENDIX

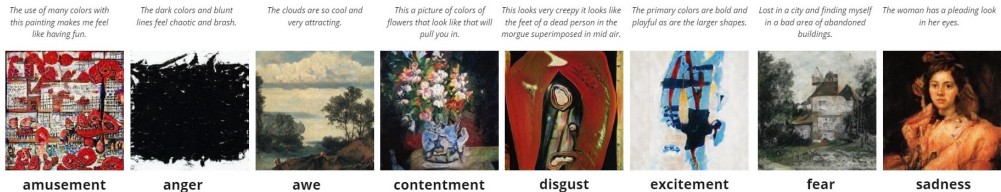

Figure 5: Example of affective captions and corresponding emotion from ArtEmis dataset and generations from HyperC-SG$^{word}$

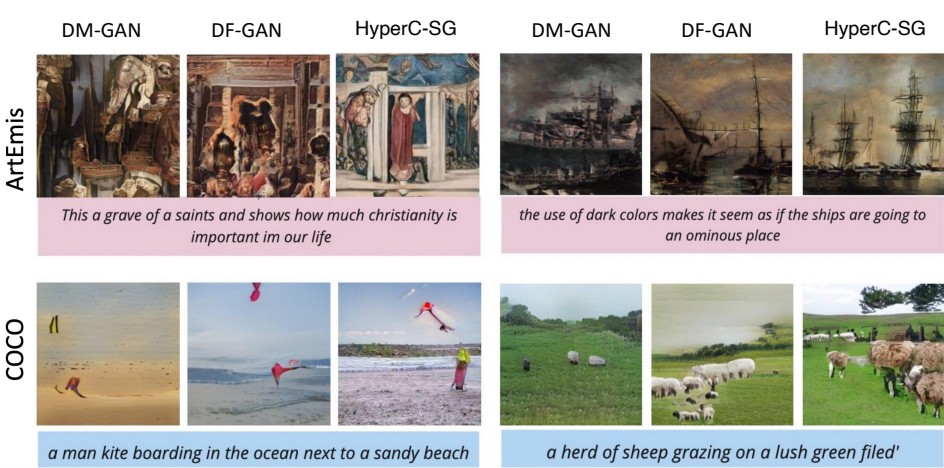

Figure 6: HyperC-SG qualitative results on ArtEmis $256^2$ (top) and on COCO $256^2$ (bottom)

## 6.1 IMPLEMENTATION DETAILS

Our models are trained with learning rate $lr = 0.0025$ with multi-gpu support on 4 NVIDIA TESLA V100 GPUs. For all experiments, we kept the batch size equal to 16 and run for 25k iterations. For COCO datasets, we followed standard splits, but we split the ArtEmis dataset into train/val/test splits in a ratio of 0.85, 0.10, 0.05. At inference, we used only test split to generate art images.

## 6.2 SENTENCE-LEVEL INFORMATION

Similar to (Xu et al., 2018; Li et al., 2019; Zhu et al., 2019; Tao et al., 2022), first we extract 256-dimensional sentence embeddings denoted as $c$ from LSTM-based pretrained text encoder. Then, we concatenate extracted embeddings and noise vector $z$ of dimension 512, and pass it through a hypernetwork $T_G(z, c)$ of the generator.

## 6.3 EFFICIENT SENTENCE LEVEL MODULATION:

To efficiently produce the conditioning tensors for sentence level modulation, we used the aforementioned factorization technique to predict the modulation mask $M$ of size $c_{out} \times c_{in} \times k_h \times k_w$ dimensions from only 4 vectors: $M = t_1 \otimes t_2 \otimes t_3 \otimes t_4$, where each $t_r$ of different dimensions $c_{out}, c_{in}, k_h, k_w$, respectively (see Figure 3):

## 6.4 WORD-LEVEL INFORMATION

In order to leverage word-level information, we extract the word embeddings from the same text encoder mentioned above. However, word embeddings have different sequence lengths and not suitable for batch processing. Therefore, the words embeddings are padded with 0s matching the max word length. Then, the padded embeddings go through hypernetworks with a single conv1x1 layers to generate style vectors of dimension $\tau \times (c_{in} + k_w + k_h)$ (See Figure 8).

```python
def tensor_modulation(weight, styles):
    """
    Performs a low-rank tensor modulation
    weight: [c_out, c_in, kw, kh] - weights of conv layer
    styles: [b, rank * (c_out + c_in + kw + kh)] - output of a hypernetwork
    """

    c_out, c_in, kw, kh = weight.shape
    b = styles.shape[0]
    rank = styles.shape[1] // (c_out + c_in + kh + kw)

    styles = styles.reshape(b, rank , c_out + c_in + kh + kw)
    # extract factors for tensor modulation
    factor1 = np.expand_dims(styles[:, :, : c_out].reshape(b, rank, c_out), [-3, -2, -1])
    factor2 = np.expand_dims(styles[:, :, c_out : c_out + c_in].reshape(b, rank, c_in), [2 ,-2, -1])
    factor3 = np.expand_dims(styles[:, :, c_out + c_in:c_out + c_in + kh].reshape(b, rank, kw), [2 ,-3, -1])
    factor4 = np.expand_dims(styles[:, :, c_out + c_in + kh:].reshape(b, rank, kh), [2 ,-3, -2])

    # obtain modulating tensor by low-rank tensor factorization
    modulating_tensor = factor1 * factor2 * factor3 * factor4 # [b, rank, c_out, c_in, kh, kw]
    modulating_tensor = modulating_tensor.sum(axis=1) / np.sqrt(rank) # [b, c_out, c_in, kh, kw]

    # Normalize the variance of 4 factors product with mean=std=1 each (assuming independence)
    modulating_tensor = modulating_tensor / np.sqrt(15)

    return modulating_tensor
```

Figure 7: Numpy-like pseudocode for core tensor modulation implementation.

| Dimension | FID | CLIP-R |
|---|---|---|
| $c_{\text{in}}$ | 18.74 | 45.13% |
| $c_{\text{out}}$ | 54.6 | 30.76% |
| $c_{\text{out}}, c_{\text{in}}$ | 21.11 | 47.86% |
| $c_{\text{in}}, k_h, k_w$ | 20.32 | 49.42% |
| $c_{\text{out}}, k_h, k_w$ | 23.49 | 43.95% |
| $c_{\text{out}}, c_{\text{in}}, k_h, k_w$ | 23.59 | 49.85% |

Table 6: Effect of different choices of modulating tensors in HyperC-SG$^{sent}$.

## 6.5 TEXT ENCODER

For text encoder, we adopt pretrained text encoder from AttnGAN. This text encoder is used in all the baselines reported in the paper. Therefore, for consistency, we also used AttnGAN text encoder which is based on a bi-directional Long Short-Term Memory (LSTM). In the bi-directional LSTM, each word corresponds to two hidden states, one for each direction. To represent the semantic meaning of a word, they concatenate its two hidden states. The last hidden states of the bi-directional LSTM are concatenated to be the global sentence vector. The hidden size of both embeddings is equal to 256.

```python
def tensor_modulation_word(weight, styles)
    """
    Performs a low-rank tensor modulation based
    on word level information
    weight: [c_out, c_in, kh, kw]
    styles: [b, c_in + kh + kw, num_words]
    """
    c_out, c_in, kw, kh = weight.shape
    b = styles.shape[0]

    styles = np.transpose(styles, (2, 1)) # [b, num_words, c_in + kh + kw]
    n_words = styles.shape[1]

    factor1 = np.expand_dims(styles[:, :, :c_in].reshape(b, n_words, c_in), [2, -2, -1])
    factor2 = np.expand_dims(styles[:, :, c_in:c_in + kw].reshape(b, n_words, kw), [2, -3, -1])
    factor3 = np.expand_dims(styles[:, :, c_in + kw:].reshape(b, n_words, kh), [2, -3, -2])

    M = factor1 * factor2 * factor3 # [b, num_words, c_in,  kh,  kw]
    M = M.reshape(b, n_words, -1) # [b, num_words, c_in x kh x kw]

    weight = np.tile(weight,(b, 1, 1, 1, 1)) # [b, c_out, c_in, kh, kw]

    weight = weight.view(b, c_out, -1) # [b, c_out, c_in x kh x kw]
    score = (weight * np.transpose(M, (2, 1))) / np.sqrt(c_out)  # [b, c_out, num_words]
    attn = softmax(score, -1)
    modulation = attn * M # [b, c_out, c_in x kh x kw]

    modulation = modulation.reshape(b, c_out, c_in, kw, kh)
    weight = weight.reshape(b, c_out, c_in, kw, kh)

    return weight * modulation
```

Figure 8: Numpy-like pseudocode for attention-based word-level tensor modulation.

## 6.6 DAMSM LOSS

DAMSM loss (Xu et al., 2018) is defined on top of Inception-v3 image model (Szegedy et al., 2016), which is used to extract image features $f \in \mathbb{R}^{768 \times 289}$ (reshaped from $768 \times 17 \times 17$). 768 is the dimension of the local feature vector, and 289 is the number of sub-regions in the image. These features are then converted to a common semantic space of text features by adding an FC layer $v = Wf$, $\quad \overline{u} = \overline{W} f$, where $v_i$ is the visual feature vector for the $i^{th}$ sub-region of the image; and $\overline{u} \in \mathbb{R}^D$ is the global vector for the whole image. We then calculate the similarity matrix for all possible pairs of words in the sentence and sub-regions in the image.

$$s = e^T v, \tag{7}$$

where $s$ is a similarity matrix between all word-region paris, $s_{i,j}$ is the dot-product similarity between the $i^{th}$ word of the sentence and the $j^{th}$ sub-region of the image. We find that it is beneficial to normalize the similarity matrix as follows

$$\overline{s}_{i,j} = \frac{\exp(s_{i,j})}{\sum_{k=0}^{T-1} \exp(s_{k,j})}. \tag{8}$$

Then, region-context vector $c_i$ is defined as a representation of the image's sub-regions related to the $i^{th}$ word of the sentence. It is computed as the weighted sum over all regional visual vectors, i.e.,

$$c_i = \sum_{j=0}^{288} \alpha_j v_j, \quad \text{where } \alpha_j = \frac{\exp(\gamma_1 \overline{s}_{i,j})}{\sum_{k=0}^{288} \exp(\gamma_1 \overline{s}_{i,k})}. \tag{9}$$

Then, the relevance between the $i^{th}$ word and the image using the cosine similarity between $c_i$ and $e_i$, i.e., $R(c_i, e_i) = (c_i^T e_i)/(||c_i|| ||e_i||)$. The *attention-driven image-text matching score* between the entire image ($q$) and the whole text description ($d$) is defined as

$$R(q, d) = \log \Big( \sum_{i=1}^{T-1} \exp(\gamma_2 R(c_i, e_i)) \Big)^{\frac{1}{\gamma_2}}, \tag{10}$$

we used the default parameters in (Xu et al., 2018).

The DAMSM loss is finally defined as

$$\mathcal{L}_{DAMSM} = \mathcal{L}_1^w + \mathcal{L}_2^w + \mathcal{L}_1^s + \mathcal{L}_2^s. \tag{11}$$

where

$$\mathcal{L}_1^w = -\sum_{i=1}^{M} \log P(d_i|q_i), \mathcal{L}_2^w = -\sum_{i=1}^{M} \log P(q_i|d_i), \tag{12}$$

where 'w' stands for "word", where $P(q_i|d_i) = \frac{\exp(\gamma_3 R(q_i, d_i))}{\sum_{j=1}^{M} \exp(\gamma_3 R(q_j, d_i))}$ is the posterior probability that sentence $d_i$ is matched with its corresponding image $q_i$. If we redefine Eq. 10 by $R(q, d) = (\overline{v}^T \overline{e})/(||\overline{v}|| ||\overline{e}||)$ and substitute it to Eq. 13 and 12, we can obtain loss functions $\mathcal{L}_1^s$ and $\mathcal{L}_2^s$ (where 's' stands for "sentence") using the sentence vector $\overline{e}$ and the global image vector $\overline{v}$. The DAMSM loss is designed to learn the attention model in a semi-supervised manner, in which the only supervision is the matching between entire images and whole sentences (a sequence of words). Similar to (Fang et al., 2015; Huang et al., 2013), for a batch of image-sentence pairs $\{(q_i, d_i)\}_{i=1}^{M}$, the posterior probability of sentence $d_i$ being matching with image $a_i$ is computed as

$$P(D_i|Q_i) = \frac{\exp(\gamma_3 R(Q_i, D_i))}{\sum_{j=1}^{M} \exp(\gamma_3 R(Q_i, D_j))}, \tag{13}$$

where $\gamma_3$ is a smoothing factor determined by experiments. In this batch of sentences, only $d_i$ matches the image $q_i$, and treat all other $M-1$ sentences as mismatching descriptions. The loss function is defined as as the negative log posterior probability that the images are matched with their corresponding text descriptions (ground truth), as shown in Eq. 12.

## 6.7 ABLATIONS FOR RANK VALUES

We experimented with different ranks to produce modulating tensor (e.g. $R = 1, 3, 5, 10$), and found that for discrete-based generator, $R = 1$ is enough to achieve good results. Increasing the rank value did not contribute to the improvement of performance but rather increased parameter sizes.

| Model | DAMSM-R ↑ | CLIP-R ↑ | FID ↓ |
|---|---|---|---|
| DM-GAN | **75.89%** | 45.07% | 16.09 |
| DF-GAN | 39.05% | 28.39% | 14.81 |
| HyperC-SG$_{DAMSM}^{sent}$ | 28.53% | 19.07% | 11.72 |
| HyperC-SG$_{DAMSM}^{word}$ | 68.53% | 21.45% | 15.02 |
| HyperCGAN$_{DAMSM}^{sent}$ | 44.15% | 26.94% | 38.66 |
| HyperCGAN$_{DAMSM}^{word}$ | 52.28% | **30.51%** | **11.00** |

Table 7: T2I Performance of HyperCGAN models on CUB (Wah et al., 2011).

| Methods | COCO $256^2$ | ArtEmis $256^2$ | CUB $256^2$ |
|---|---|---|---|
| Nearest | 30.86 | 26.87 | 17.69 |
| Bilinear | 29.84 | 28.06 | 16.84 |
| Bicubic | 28.73 | 26.52 | 16.18 |
| HyperCGAN | **27.61** | **23.78** | **15.42** |

Table 8: Super-resolution Synthesis comparison on FID scores. In this setting, we trained models on downsampled $128^2$ images and generate $256^2$ resolution images without changing architecture or finetuning.

### 6.8 ADDITIONAL RESULTS ON CUB DATASET

We performed additional experiments where we trained our models on CUB dataset and compared to recent baselines (DF-GAN, DM-GAN). In Table 7, the results indicate that our models achieve the best FID and comparable CLIP-R scores.

### 6.9 OUT-OF-THE-BOX SUPERRESOLUTION GENERATION

In this setting, we conduct out-of-the-box generation in a training efficiency manner naturally inherited from INR-based model. We first train our model on downsampled images ($128 \times 128$) and perform $256 \times 256$ generation during inference either with classical interpolation methods or taking advantage of INR-based model's merits. This means our model can generate higher resolution images without modifying any architecture or finetuning, by just adjusting to a more denser coordinate grid. Table 8 show that our model outperforms standard upsampling techniques on all datasets. See Figure 9 for qualitative results.

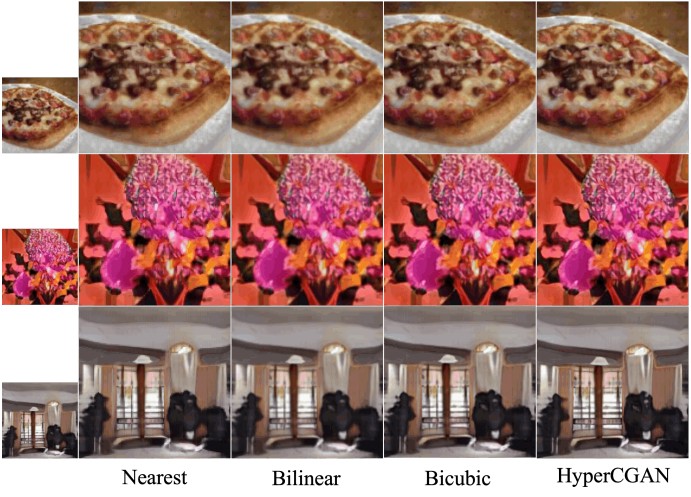

| Nearest | Bilinear | Bicubic | HyperCGAN |

Figure 9: Qualitative comparison between classical interpolation techniques and our model.

## 6.10 Generating High Resolution Images (COCO)

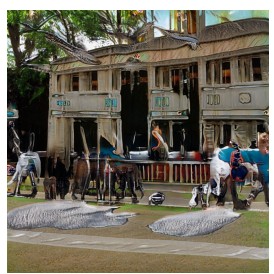 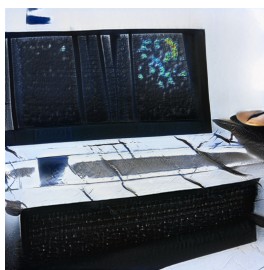 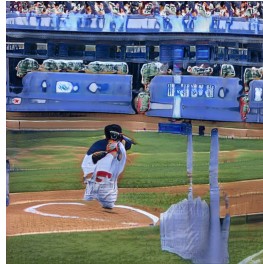

a house being built with lots of wood  ·  a laptop computer sits on a computer desk next to a mouse  ·  a batter backs up as the ball is thrown

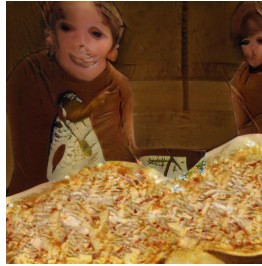 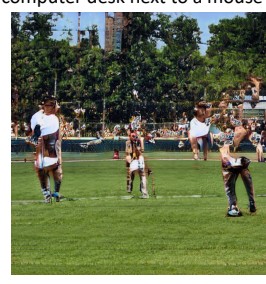 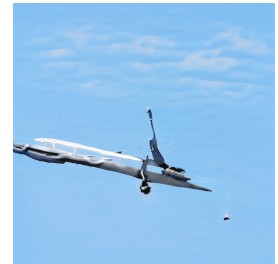

a man holding a fully topped pizza in front of the camera  ·  soccer players are running after the ball together  ·  a plane is flying high in the very cloudy sky

Figure 10: High-resolution generations (1024x1024) from our HyperC-SGs trained on COCO.

| Model | COCO $256^2$ | ArtEmis $256^2$ | CUB $256^2$ |
|---|---|---|---|
| AttnGAN (Xu et al., 2018) | 81.52% | 78.68% | 67.82% |
| ControlGAN (Li et al., 2019) | 82.43% | 78.75% | 69.33% |
| DM-GAN (Zhu et al., 2019) | **88.56**% | **93.54**% | **75.89**% |
| XMC-GAN (Zhang et al., 2021) | 69.75% | 34.37% | 47.22% |
| DF-GAN (Tao et al., 2022) | 55.85% | 52.38% | 39.05% |
| HyperCGAN$_{\text{DAMSM}}^{word}$ | 70.24% | 22.04% | 52.28% |
| HyperCGAN$_{\text{CLIP}}^{word}$ | 53.06% | 34.48% | 27.15% |
| Real Images | 22.22% | 23.87% | 20.00% |

Table 9: **DAMSM-R Metric Limitations (comparison among T2I models).** Previous works heavily skew to DAMSM-R compared to real images for COCO (Zhang & Schomaker, 2020), CUB (Park et al., 2021) and ArtEmis.

