# OpenReview forum: "Adversarial Text to Continuous Image Generation"
_ICLR.cc/2023/Conference — Submitted to ICLR 2023_

### Official Review · Reviewer_xx7K · 2022-10-25

**Confidence:** 4
**Correctness:** 3
**Technical Novelty And Significance:** 2
**Empirical Novelty And Significance:** 2
**Recommendation:** 5

**Clarity, Quality, Novelty And Reproducibility:**

The proposed method proved that INR-GAN-based works can also be extended for the text-to-image generation task.
However, the traits that make INR-GAN-based works tempting such as outpainting an original image is not good enough, especially considering current diffusion-based models' ability to generate quite impressive results not only in outpainting but also in super-resolution images.
Moreover, the proposed method of using word-level attention mechanism is somewhat incremental, and its efficacy is not really clear as the remark I made regrading the ablation study.

**Strength And Weaknesses:**

Strengths
- The proposed method successfully extend INR-GAN for the text-to-image generation task.
- Improved scores on some quantitative evaluation metrics compared to prior GAN-based works.


Weaknesses
- The reason of choosing INR-based generator would be to generate images free from the constraint on the image resolution. But the results that promote this aspect is insufficient, and in fact, it does not seem to be helpful for generating realistic images. For example, the two images in the first column in Fig 1. show that they are merely mirroring the edges of the original images, which hinder their realism. It is not also analyzed in quantitative manner.
- The content in Related Work part is poor. I understand that this work is based on GAN, and might not be fair to compare with current diffusion models, but it would be better to mention those works and differentiate this work describing what factors make this work more beneficial compared to them.
- I have an impression that the DAMSM regularizer has a larger influence than the proposed method in improving the image quality from Table 2. Maybe the order should be reconfigured if the authors want readers to focus on the efficacy of using word-level modulating mechanism.
- The authors did not show how much computational burden can be reduced by using the canonical polyadic decomposition method.



**Summary Of The Paper:**

This work proposes INR-GAN-based text-to-image generation method, where model weights are modulated with the additional module called hypernetwork and RGB value is predicted given pixel coordinate as input.
To make a model to reflect the textual condition better, the authors propose a new method of generating modulating weights by incorporating word-level predictions.
Specifically, they first compute word-level modulating weights and then merge these values based on their similarities between the weights of the model.
Further, to reduce the computational burden, the authors propose to use the canonical polyadic decomposition to decompose the tensor to predict.

**Summary Of The Review:**

Extending INR-GAN in the text-to-image generation task is an interesting approach, and the proposed method show improved results compared to GAN-based prior methods.
However, the demonstrated improvements are somewhat limiting both considering its lack of ability to extend images retaining the realism of the original images and
research on prior text-to-image generation works.

---

> ### Author Response · Authors · 2022-11-17
> **Part 1: Response to Reviewer xx7K**
>
> We thank Reviewer xx7K for the valuable feedback. We address below the raised concerns.
>
> >The content in Related Work part is poor. I understand that this work is based on GAN, and might not be fair to compare with current diffusion models, but it would be better to mention those works and differentiate this work describing what factors make this work more beneficial compared to them.
>
> Thanks for pinpointing this deficiency in Related Work section. We agree that discussing diffiusion models would benefit our paper. We updated the Related Work section with works based on Non-GAN based approaches like invertible-neural networks, diffusion-based models like, DALL-E2, IMAGEN, and Stable Diffusion to bring more clarity to our manuscript.”  They indeed introduce a significant advance in the field and the results are amazing. However, these methods cannot be directly compared with our work since they have a huge number of parameters and are trained on a massive amount of data”. The main idea behind these methods is that they do denoising operations conditioned on text. The nice property of diffusion-based methods is that they do not suffer from mode collapse, but their generative cost is higher than GAN-based approaches. Changes are highlighted with orange color in the Related work section (Page 3).
>
>
> >The authors did not show how much computational burden can be reduced by using the canonical polyadic decomposition method.
>
> Thanks for the comment. We have already provided an example of how low-rank factorization reduces the number of parameters in the Extreme Modulating Tensor Factorization section. For convenient reference, we quote this statement from our paper (page 6):
>
> **“Producing a full-rank tensor Mℓ for each block l is memory-intensive and infeasible even for modestly sized architectures. For example, if the hidden layer size of our hypernetwork is of size dh = 512 and the convolutional weight tensor at layer ℓ is of dimensionality do = cout × cin × kh × kw = 512 × 512 × 3 × 3 ≈ 2.4 million, then the output weight matrix in the hypernetwork will be of size do × dh = 1.2 billion. “**
>
> To validate this claim experimentally, we have tried to generate the full-rank tensor using hypernetworks without CP tensor decomposition. As expected, this lead to Out-Of-Memory error in V100 gpus with 32 GB memory (the configuration with the largest available GPU memory from NVIDIA Volta architecture)
>
>
> **“To overcome this issue, we propose factorizing the modulating tensor with an extreme low-rank tensor decomposition for learning efficiency…”**
> If we instead generate separately low-rank factors and build modulating tensor out of the factors do = cout + cin + kh + kh = 512 + 512 + 3 + 3 = 1030. So the output weight matrix in the hypernetwork will be of size do × dh = 527360 which leads to 99.95% decrease in parameter size of hypernetworks.

---

> > ### Author Response · Authors · 2022-11-17
> > **Part 2: Response to Reviewer xx7K**
> >
> > >I have an impression that the DAMSM regularizer has a larger influence than the proposed method in improving the image quality from Table 2. Maybe the order should be reconfigured if the authors want readers to focus on the efficacy of using word-level modulating mechanisms.
> >
> > Thanks for your suggestions. Regularizers (DAMSM or CLIP) are standard practice to improve the image-text semantic consistency and they are orthogonal to our proposed WhAtt mechanism; see Condition INR-GAN + DAMSM (config C), HyperCGAN_sentence + DAMSM (config F), HyperCGAN_word + DAMSM (config L) below.  Also, all the previous works  (e.g.AttnGAN, ControlGAN, DM-GAN) rely on some sort of auxiliary losses/regularizers to improve image-semantic consistency. Since this naturally leads to image improvement. We first show how naive conditioning and HyperCGAN with sentence conditioning behave without and with regularizers and then show how our HyperCGAN with the proposed hypernetwork-based word-level attention mechanism WhAtt achieves superior results when it is guided by regularizers.  For example, our WhAtt mechanism enables the model to achieve 37.23% and 25.39 in terms of CLIP-R and FID scores, respectively. If we combine the model with contrastive regularizers, we achieve R-scores more than 60%. For convenience, we compiled the methods that uses DAMSM regularizers below to clarify the contribution of the  WhAtt mechanism.
> >
> > | Method                                | CLIP-R | FID   |
> > |---------------------------------------|--------|-------|
> > | AttnGAN   (uses DAMSM)   | 29.31% | 35.49 |
> > | ControlGAN    (uses DAMSM)          | 24.96% | 34.52 |
> > | DM-GAN      (uses DAMSM)                       | 40.31% | 32.64 |
> > | Condition INR-GAN + DAMSM (config C)  | 57.11% | 18.51 |
> > | HyperCGAN_sentence + DAMSM (config F) | 51.34% | 29.66 |
> > | HyperCGAN_word + DAMSM (config L)     | 64.14% | 18.58 |
> >
> > Specifically, DAMSM combined with WhAtt mechanism boosts scores 64.14% and 18.58 in DAMSM-R and FID scores, respectively. For convenience, in the table below, we report the performance of all the baselines and SOTA methods that uses DAMSM regulirizer in terms of FID and CLIP-R scores. In general, we can observe that our WhAtt attention mechanism is crucial to guide the model in achieving the best results (last row).
> >
> >
> > ---------------------------------------
> >
> >
> > The remaining concerns will be addressed in follow-up responses:
> > >The reason of choosing INR-based generator would be to generate images free from the constraint on the image resolution. But the results that promote this aspect is insufficient, and in fact, it does not seem to be helpful for generating realistic images. For example, the two images in the first column in Fig 1. show that they are merely mirroring the edges of the original images, which hinder their realism. It is not also analyzed in a quantitative manner.

---

> > > ### Author Response · Authors · 2022-11-19
> > > **Part 2: Response to Reviewer xx7K**
> > >
> > > >The reason for choosing an INR-based generator would be to generate images free from the constraint on the image resolution. But the results that promote this aspect is insufficient, and in fact, it does not seem to be helpful for generating realistic images. For example, the two images in the first column in Fig 1. show that they are merely mirroring the edges of the original images, which hinder their realism. It is not also analyzed in a quantitative manner.
> > >
> > > Thanks for your concern. Even though that some extrapolated regions in our generated examples are not close to realism,  we need to kindly remind you that extrapolation is being done without any special training. We provide more examples in the accompanying website [1]. As it can be observed, most of the extrapolated areas do not merely copy/mirror the edges of original training resolutions (region inside the red rectangles): Please, from provided examples in the website observe how naturally extended:
> > > The tails of the birds or the branches of trees (see CUB examples)
> > > The faces/silhouettes of figures (see ArtEmis examples)
> > > The legs of a baseball player, the bus, and the corner of plates (COCO example)
> > >
> > > Also, quantitatively, we reported the result related to extrapolated generations from user studies which we believe the best way to assess extrapolated regions. We showed:
> > > 1. whether the extended area in the generations is meaningful (Table 3)
> > > 2. whether the extended area makes the image more aligned with the text description. (Table 4)
> > >
> > > Results suggest that 68.8% of responses indicate that the out-of-the-region extrapolation is meaningful while 20.8% of them say it is not for COCO. For ArtEmis, the results are higher; 75.6% of responses are in favor of meaningful; meanwhile, 16.4% of them show the opposite. In Table 4, more than 65% of responses suggested that the alignment between the image and text description improved or remained the same for both COCO and ArtEmis.
> > >
> > >
> > > [1] https://hypercgan-website.s3.amazonaws.com/index.html

---

### Official Review · Reviewer_3nfM · 2022-10-25

**Confidence:** 5
**Correctness:** 3
**Technical Novelty And Significance:** 3
**Empirical Novelty And Significance:** 3
**Recommendation:** 5

**Clarity, Quality, Novelty And Reproducibility:**

**Clarity:** The approach section can be made clearer to better convey the information flow of the WhAtt component of the framework. In particular, the way the word embeddings are used to generate the final modulation vector at each layer is unclear.

**Quality:** The quality of writing and results is adequate. Efforts have been taken to report both qualitative and quantitative results. The related work section could use some discussion about diffusion based pipelines for text driven image generation.

**Novelty:** The approach is sufficiently novel. The authors propose a hypernetwork based design for text driven image synthesis using an implicit representation . Although similar ideas have been explore in Scene Representation Networks (Sitzmann et al.) this work provides important extension of the framework to the text guided case.

**Reproducibility:** Due to lack of code and some unclear technical details of the WhAtt component, the method maybe hard to reproduce.

**Strength And Weaknesses:**

## Strength

1. **Writing:** The paper is well written with all components explained in adequate detail.

2. **User study:** The user study performed demonstrates the effectiveness of the approach.

3. **Comparison:** Adequate comparisons are made both with respect to baseline approaches and various dataset highlighting the usefulness of the approach.

4. **Parameter efficiency:** The proposed approach provides a high degree of parameter efficiency compared to multistage baseline approaches, while also giving the ability for extrapolation and superresolution for free. This is particularly useful in not being tied to a specific spatial resolution while generating an image.

## Weakness

1. **Related work:** Any manuscript on text to image generation would be incomplete without the inclusion of the recent large text to image diffusion models.  The related work only tackles GAN based text to image generation without referencing any of the advances in diffusion models. Although each method has distinct advantages and disadvantages, the related work section would benefit from a more in depth treatment of this line of work.
2. **Approach:** The section regarding how the individual word level embeddings are used to modulate the network weights is unclear. Particularly, a more detailed description of the WhAtt component would be instructive.
2. **Experimental analysis:** There are certain sections of the experiment section which is unclear, Particularly,
>a) It is unclear what the DAMSM-R score represents. Particularly, how do we interpret the percentage score here?
>b) The ablation study demonstrates the need of some of the losses but it does not make the need for a conditional discriminator clear. In particular, why does the discriminator also need to be modulated with a hypernetwork ? Any isights regarding this would be helpful.
4. **Reproducibility:** Some aspects of the WhAtt component are unclear, making it hard to reimplement and reproduce the framework.


**Summary Of The Paper:**

The authors propose a framework for text to image generation using an implicit image generation model. The text conditioning is achieved through means of a novel hypernetwork inspired architecture that modulates the weights of the generator and discriminator network based on the text prompts. State of the art results are demonstrated on 3 datasets.


**Summary Of The Review:**


Although the authors present a reasonably novel approach for text to image synthesis, this work would greatly benefit from an adequate treatment of the diffusion models literature and some more details regarding the word level hyper network design.

---

> ### Author Response · Authors · 2022-11-17
> **Part 1: Response to Reviewer 3nfM**
>
> We thank Reviewer 3nfM for the valuable feedback. We here address the comments and will incorporate all the feedback.
>
> >Related work: Any manuscript on text-to-image generation would be incomplete without the inclusion of the recent large text to image diffusion models. The related work only tackles GAN-based text-to-image generation without referencing any of the advances in diffusion models. Although each method has distinct advantages and disadvantages, the related work section would benefit from a more in-depth treatment of this line of work.
>
> Thanks for pinpointing this deficiency in Related Work section. We agree that discussing diffiusion models would benefit our paper. We updated the Related Work section with works based on Non-GAN based approaches like invertible-neural network, diffusion based models like, DALL-E2, IMAGEN, Stable Diffusion to bring more clarity to our manuscript.”  They indeed introduce a significant advance in the field and the results are amazing. However, these methods cannot be directly compared with our work since they have a huge number of parameters and are trained on a massive amount of data”. The main idea behind these methods is that they do denoising operations conditioned on text. The nice property of diffusion-based methods is that they do not suffer from mode collapse, but their generative cost is higher than GAN-based approaches. Changes are highlighted with orange color in Related work section (Page 3).
>
> >Approach: The section regarding how the individual word level embeddings are used to modulate the network weights is unclear. Particularly, a more detailed description of the WhAtt component would be instructive.
>
> Thanks for the suggestion. Kindly let us go over this process step by step and how it is reflected in the paper (Page 6). These steps are now included as Numpy-like pseudocode in Figure 8 in the Appendix for further clarity; the code is also attached in this update: https://drive.google.com/file/d/1HDDlOll_0Vrd-y7vXj2-ph96jJkJ7K8t/view?usp=sharing
> (location src/training/tensor_mod.py)
>
> 1. First we obtain word embeddings from the text encoder using bidirectional LSTM following [AttnGAN, ControlGAN, DM-GAN] papers; this text encoder is the one also used in the DAMSM module.
> 2. These word embeddings have the dimensionality of Ω×d where Ω denotes sequence length  (i.e., the number of words) and d is an embedding size.
> 3. These word embeddings are passed through a different HyperNetwork which consists of a single Conv1x1 layer for each layer $\ell$. From these each conv1x1-based hypernetwork at layer $\ell$, we obtain a different tensor  $\mathcal{T}^\ell$ of size Ω×($c_{in} + k_h + k_w$). This tensor  $\mathcal{T}^\ell$  is composed of  Ω number of different vectors $\mathbf{v}^i$ of size $c_{in} + k_h + k_w$ corresponding to the i-th word.
> 4. Then, tensor  $\mathcal{T}^\ell$ is decomposed into 3 parts; basically,  we “slice” each vector $\mathbf{v}^i$ to get 3 low-rank factors $\mathbf{v}^i_\mathrm{in}$, $\mathbf{v}^i_h$, $\mathbf{v}^i_w$ of dimensions $c_\text{in}$, $k_h$, $k_w$, respectively.
> 5. Out of these factors we build a tensor $Q^\ell$ using outer product operation according to Eq 3.
> 6. , and use the attention mechanism to obtain modulating tensor $M_w^\ell$ which takes weight matrix $W^\ell$  as a query and $Q^\ell$  as a key input. (see Eq 4)
> 7. This tensor $M_w^\ell$   is further used to modulate weight matrix  $W^\ell$  at layer $\ell$ with elementwise multiplication (see Eq 6)
> Changes are reflected in orange color in Page 5, 6.
>
> The implementation of WhAtt attention can be found in the code  (src/training/tensor_mod.py).
>
> >Experimental analysis: There are certain sections of the experiment section which is unclear, Particularly,
> a) It is unclear what the DAMSM-R score represents. Particularly, how do we interpret the percentage score here?
>
> In order to evaluate to what extent the generated images are aligned with the given input text descriptions, AttnGAN and the following works (ControlGAN, DM-GAN) utilize pre-trained Deep Attentional Multimodal Similarity Model (DAMSM) to compute the retrieval score. We followed the same protocol. DAMSM consists of a text and image encoder which learns to map subregions of the image and words of the sentence to a common semantic space. During the evaluation, compute cosine similarities between the global vectors of generated images and the global vectors of text descriptions from encoders of DAMSM are used to rank candidate text descriptions for each generated image in descending similarity and find the top R relevant descriptions for computing the R-precision. To put it simply, the percentage means how many times the input text description appeared at TOP-1 among candidate descriptions after ranking.

---

> > ### Author Response · Authors · 2022-11-17
> > **Part 2: Response to Reviewer 3nfM**
> >
> > >Reproducibility: Some aspects of the WhAtt component are unclear, making it hard to reimplement and reproduce the framework.
> >
> > To make reproducibility clear, we are sharing the code through this anonymous link https://drive.google.com/file/d/1HDDlOll_0Vrd-y7vXj2-ph96jJkJ7K8t/view?usp=sharing
> > For WhAtt attention modulation, kindly refer to (location src/training/tensor_mod.py) and  Figure 8 in the Appendix.
> >
> > ---------------------------------------
> >
> > The concerns below will be addressed in follow-up responses:
> >
> > >The ablation study demonstrates the need of some of the losses but it does not make the need for a conditional discriminator clear. In particular, why does the discriminator also need to be modulated with a hypernetwork ? Any isights regarding this would be helpful.

---

> > > ### Author Response · Authors · 2022-11-19
> > > **Part 2: Response to Reviewer 3nfM**
> > >
> > > >The ablation study demonstrates the need of some of the losses but it does not make the need for a conditional discriminator clear. In particular, why does the discriminator also need to be modulated with a hypernetwork ? Any isights regarding this would be helpful.
> > >
> > > Instead of WhAtt modulation in the discriminator, we conditioned the final projection head of the original StyleGAN-2 on the average of word embeddings. The performance of this model reported  in the table below:
> > >
> > > | HyperCGAN^${word}$ with StyleGAN2 Discriminator | 26.18%  | 55.13 |
> > > |----------------------------------------------|---------|-------|
> > > | HyperCGAN$^{word}$ with StyleGAN2 Discriminator + DAMSM regularizer                            | 45.36%  | 35.21 |
> > > | HyperCGAN$^{word}$ with StyleGAN2 Discriminator+ CLIP regularizer                             | 33.18%  | 47.82 |
> > > | HyperCGAN$^{word}_{DAMSM}$ (ours)                  | 64.14%  | 18.58 |
> > >
> > >
> > > As you can see, the WhAtt attention effectively helps the underlying decoder to learn image generation from text and improves results both in terms of FID and CLIP-R retrieval scores by parametrizing the backbone through modulation mechanism. Compared to StyleGAN-2 Discriminator, the proposed hypermodulated discriminator with WhAtt mechanism achieves superior performance 45.36% vs 64.14% in CLIP-R, and 35.21 vs 18.58 in terms of FID, respectively.

---

### Official Review · Reviewer_opUk · 2022-10-26

**Confidence:** 3
**Correctness:** 4
**Technical Novelty And Significance:** 2
**Empirical Novelty And Significance:** 3
**Recommendation:** 6

**Clarity, Quality, Novelty And Reproducibility:**

The paper is very well explained and easy to read.
In terms of reproducibility I dont see any error bars (runs with multiple seeds). This is important for GAN based generative models as they can have a lot of variability from one seed to another.

The novelty of the paper is rather very limited and a straight forward extension of a previous work INR-GAN which is an unconditional INR based GAN.

**Strength And Weaknesses:**

The approach is interesting as it achives similar or better performance  as other SOTA methods with much less parameters.
Have you done any running time experiments to see how the training and generation time compares with SOTA. The experiments are extensive and done on multiple datasets.
The only point of concern is the limited novelty in terms of the actual approach it self.
I dont see any error bars (runs with multiple seeds). This is important for GAN based generative models as they can have a lot of variability from one seed to another.

**Summary Of The Paper:**

The authors present a method for Implicit Neural Representations conditional continuous image generation. Basically the authors augment  INR-GAN, which is a hypernetwork based generative adversarial network for image generation, to include text embeddings. This is doen either by concatenation or by using an additional hypernetwork for the text embeddings.

**Summary Of The Review:**

Text to image generation has gained a lot of popularity in recent months and it would be interesting to benchmark the current approach
 (which can run with little amount of parameters) with ViT based and diffusion based approaches.  I do believe the paper has merits in application but the theoretical novelty is very limited.

---

> ### Author Response · Authors · 2022-11-17
> **Response to Reviewer opUk**
>
> >I don't see any error bars (runs with multiple seeds). This is important for GAN-based generative models as they can have a lot of variability from one seed to another.
>
> Thanks for your comment. We started from the INR-GAN code base and we only added our conditioning modules with the default hyper-parameters including the seed, and we ablated all the components of our method in the ablation table (Table1). For fair comparison, we follow the standard practice in reporting T2I performance (e.g.  [2, 3, 4, 5, 6]) . None of these GAN papers which train medium-to-large-sized models (except for BigGAN [1]) shows error bars literally.
>
> Multiple runs are also not environmentally friendly since the power consumption used to train the models will contribute to carbon emissions [7]. We can estimate this through the following equation. Power consumption x Time x Carbon Produced Based on the Local Power Grid.Our experiments require 4 v100 32GB gpus and requires 384 GPU hours in total to converge. Then, one run produces
> 300W x 384h = 115.2 kWh x 0.3 kg eq. CO2/kWh = 34.56 kg eq. CO2
> This means that 139 km driven by an average ICE car or 17.3 kgs of coal burned
>
>
> [1] https://arxiv.org/pdf/1809.11096.pdf
> [2] https://arxiv.org/pdf/2101.04702.pdf
> [3] https://arxiv.org/pdf/1812.04948.pdf
> [4] https://arxiv.org/pdf/2011.12026.pdf
> [5]https://openaccess.thecvf.com/content_CVPR_2020/papers/Schonfeld_A_U-Net_Based_Discriminator_for_Generative_Adversarial_Networks_CVPR_2020_paper.pdf
> [6] https://nvlabs-fi-cdn.nvidia.com/stylegan3/stylegan3-paper.pdf
> [7] https://arxiv.org/pdf/1910.09700.pdf
>
>
> >The novelty of the paper is rather very limited and a straightforward extension of a previous work INR-GAN which is an unconditional INR-based GAN.
>
> We respectfully disagree. We showed in our experiments that straightforward adaptation of INR-GAN to a conditional setting is not straightforward and introducing word-level hypernetworks as well as our introduced WhAtt attention mechanism is necessary to achieve good performance. We design a set of complex baselines (e.g.. Table 1, rows 2-7), with standard sentence conditioning (rows 2-4) and an improved version with a hyper-modulated discriminator (rows 5–7).
>
> We observe that sentence conditioning (Table 1 rows 2-4) achieves 57.11% and 49.71% CLIP-R (which evaluates the image-text semantic consistency). In contrast, our word-level WhAtt mechanism provides more fine-grained information and boosts the CLIP-R score to 64.14%(+ 7.03%) and 63.99% (+14.28) respectively (Table 1 row 8-10), which proves the significance of using the WhAtt mechanism for semantic consistency in the T2I setting. We also show the generalization of the WhAtt attention to boosting performance on discrete decoders as well (Table 2).
>
> For reference, we detail the steps of the WhAtt mechanism below steps in the Numpy-like pseudocode in Figure 8 in the Appendix for further clarity; the code is also attached in this update: https://drive.google.com/file/d/1HDDlOll_0Vrd-y7vXj2-ph96jJkJ7K8t/view?usp=sharing
> (location src/training/tensor_mod.py). We are not aware of a mechanism that is similar to the proposed one in terms of both it is designed and how it modulates the weights of the generator and discriminator.
>
> 1. First we obtain word embeddings from the text encoder using bidirectional LSTM following [AttnGAN, ControlGAN, DM-GAN] papers; this text encoder is the one also used in the DAMSM module.
> 2. These word embeddings have the dimensionality of Ω×d where Ω denotes sequence length  (i.e., the number of words) and d is an embedding size.
> 3. These word embeddings are passed through a different HyperNetwork which consists of a single Conv1x1 layer for each layer $\ell$. From these each conv1x1-based hypernetwork at layer $\ell$, we obtain a different tensor  $\mathcal{T}^\ell$ of size Ω×($c_{in} + k_h + k_w$). This tensor  $\mathcal{T}^\ell$  is composed of  Ω number of different vectors $\mathbf{v}^i$ of size $c_{in} + k_h + k_w$ corresponding to the i-th word.
> 4. Then, tensor  $\mathcal{T}^\ell$ is decomposed into 3 parts; basically,  we “slice” each vector $\mathbf{v}^i$ to get 3 low-rank factors $\mathbf{v}^i_\mathrm{in}$, $\mathbf{v}^i_h$, $\mathbf{v}^i_w$ of dimensions $c_\text{in}$, $k_h$, $k_w$, respectively.
> 5. Out of these factors we build a tensor $Q^\ell$ using outer product operation according to Eq 3.
> 6. , and use the attention mechanism to obtain modulating tensor $M_w^\ell$ which takes weight matrix $W^\ell$  as a query and $Q^\ell$  as a key input. (see Eq 4)
> 7. This tensor $M_w^\ell$   is further used to modulate weight matrix  $W^\ell$  at layer $\ell$ with elementwise multiplication (see Eq 6)
> Changes are reflected in orange color in Page 5, 6.
>
>
> ---------------------------------------
>
> The concerns below will be addressed in follow-up responses:
> > Have you done any running time experiments to see how the training and generation time compares with SOTA.

---

### Official Review · Reviewer_cHep · 2022-10-26

**Confidence:** 3
**Correctness:** 3
**Technical Novelty And Significance:** 2
**Empirical Novelty And Significance:** 2
**Recommendation:** 3

**Clarity, Quality, Novelty And Reproducibility:**

In terms of clarity and novelty, I've found some issues on the contribution of this work. Please refer to my detailed comments above.

For reproducibility, I haven't found major issues to implement the proposed method. I expect all the number in this manuscript would be reproducible.

**Strength And Weaknesses:**

Strengths:
* The motivation and details of HyperCGAN are reasonable. To minimize the cost of INR, using CP decomposition is also appealing.
* Experiments show that the proposed approach is able to extrapolate the outside of the input boundary.

Weaknesses:
* In my opinion, the novelty of HyperCGAN is quite limited, since this could be interpreted as a fairly straightforward extension to INR-GAN for processing the text condition. Obviously, injecting the text condition to INR-GAN brings many practical advantages, but more technical contribution would be required.
* The out-painting quality of this method is quite worse than SOTA methods, including infinity GAN and NUWA-infinity:
  * https://hubert0527.github.io/infinityGAN/
  * https://arxiv.org/abs/2207.09814 (NUWA-infinity)
* Clarity needs to be improved. For instance, I failed to see why introducing another T2I benchmark improves the contribution of this work.

**Detailed comments**

I’m not sure whether the proposed T2I benchmark adds the value of this work or not.  In most cases, establishing a new benchmark contributes to the research community significantly. However, in this work, I failed to see the connection of the new benchmark and proposed method. Could you elaborate why the existing datasets are not good enough to evaluate the proposed method?

In my understanding, WhAtt does not take any sequence-level information. In CLIP, the sequence-level and sub-word level information are both useful representations, so it might be better to make use of both information in modulating the weights.

Table 2 shows DAMSM regularizer performs better than CLIP. It seems to be quite unusual, since the representation power of DAMSM would be weaker than CLIP for most cases. Could you elaborate more why DAMSM is better than CLIP?

I’ve found that the comparison to SOTA in the T2I task is quite misleading, since the proposed method is not compared to real SOTA methods. In addition, the authors argue that the proposed method outperforms DALL-E 1, but DALL-E 1 reports the performance of zero-shot evaluation, which is not the case of this work.

**Summary Of The Paper:**

This work tackles to solve the text-to-continuous image generation task (T2CI) by introducing a hypernetwork that modulates the weights of discriminator and generator based on the text condition. The proposed architecture can be considered as an extension of INR-GAN, making the modulator take the text condition. In specific, the authors have proposed two types of modulator to handle sentence-level or world-level information. Experiments on small-scale datasets have shown that the proposed approach is better than existing INR-based generative models in the T2CI task.

**Summary Of The Review:**

I’m leaning towards rejection, since the novelty of the proposed method is somewhat limited, and insufficient empirical justification makes it difficult to evaluate the potential of the method.

---

> ### Author Response · Authors · 2022-11-17
> **Part 1: Response to Reviewer cHep**
>
> >In my opinion, the novelty of HyperCGAN is quite limited, since this could be interpreted as a fairly straightforward extension to INR-GAN for processing the text condition. Obviously, injecting the text condition to INR-GAN brings many practical advantages, but more technical contribution would be required.
>
> We respectfully disagree. We showed in our experiments that straighforward adaptation of INR-GAN to a conditional setting is not working well and introducing word-level hypernetworks as well as our introduced WhAtt attention mechanism, we found necessary to achieve good performance. We design a set of complex baselines (e.g.. Table 1, rows 2-7), with standard sentence conditioning (rows 2-4) and an improved version with a hyper-modulated discriminator (rows 5–7).
>
> We can observe that sentence conditioning (Table 1 row 2-4) achieves 57.11% and 49.71% CLIP-R (which evaluates the image-text semantic consistency). In contrast, our word-level WhAtt mechanism provides more fine-grained information and boosts the CLIP-R score to 64.14%(**+ 7.03%**) and 63.99% (**+14.28%**) respectively (Table 1 row 8-10), which proves the significance of using the WhAtt mechanism for semantic consistency in the T2CI setting. We also show the generalization of the WhAtt attention to boosting performance on discrete decoders as well (Table 2).
>
> >“The out-painting quality of this method is quite worse than SOTA methods, including infinity GAN and NUWA-infinity:”
> >https://hubert0527.github.io/infinityGAN/
> >https://arxiv.org/abs/2207.09814 (NUWA-infinity)
>
> Thanks for pointing out these works; The scope of our work is different. We characterize the differences below:
> 1. Our focus is not out-painting but rather on enabling Text-to-continuous-Image generation with semantic consistency. By enabling continuous GAN models for T2I, we show that our method retains extrapolation abilities while conditioning on the text. We used a standard text-2-image dataset including COCO and CUB.
> 2. In HyperCGAN, out-painting (out-of-the-box extrapolation) comes naturally happens **without any special training**, which is not the case in NUWA-infinity.  By enabling continuous GAN models for T2I, we show that our method retains extrapolation abilities with visual semantic consistency.
> 3. Unfair comparison, our whole model size is 65M with the Decoder being of size 36M, while the NUWA-infinity decoder size is only **809M** as reported in Table 7 (b) [1]. This is **12.44** times larger than the size of our whole model, and about  **22.47** times large than our Decoder model.
> 4. InfinityGAN and NUWA-infinity scope is different: InfinityGAN is evaluated on Flickr-Landscape and a scenery-related subset of Place365 and Flickr-Scenery (different from Flickr-Landscape) which are less object-centric. Similarly,  to evaluate the ability of NUWA-infinity on outpainting, authors build custom datasets (RQF, LHQC, LHQ-V) that are mostly related to scene-centric images. Of course, the Riverside of Qingming Festival (RQF) dataset has more detailed compositions of people, objects, and scenes. Yet,  this dataset contains 128 images and is composed of parts of the artwork “Along the River During the Qingming Festival” drawn by Qiu Ying. It is still not that diverse since samples resemble each other in style and content. Lastly, the text-to-image generation ability of NUWA-infinity is focusing on scene-related generation.
>
> [1] https://arxiv.org/abs/2207.09814 (NUWA-infinity)

---

> > ### Author Response · Authors · 2022-11-17
> > **Part 2: Response to Reviewer cHep**
> >
> > >“Clarity needs to be improved. For instance, I failed to see why introducing another T2I benchmark improves the contribution of this work. I’m not sure whether the proposed T2I benchmark adds the value of this work or not. In most cases, establishing a new benchmark contributes to the research community significantly. However, in this work, I failed to see the connection of the new benchmark and proposed method. Could you elaborate why the existing datasets are not good enough to evaluate the proposed method?”
> >
> > Current standard benchmarks for T2I (e.g. MS-COCO, CUB) are mainly centered around real images and contain objective language (merely describing what is depicted in the image). Contrary to these benchmarks, ArtEmis dataset is based on 81k artworks from the WikiArt dataset and the associated language is affective. The dataset was collected such that the associated text is an explanation of a constructed emotional experiments, such as excitement. Hence, studying text-to-image generation with this dataset helps us explore the ability of these models to generate not only artworks but also the ability to understand affective language. Successful T2I models need to learn how to generate visually artistic images from affective captions. So, we introduced the evaluation of our proposed and existing  methods on this benchmark to see how it handles affective language to generate visual stimuli (as a step generating images that may construct emotional experiences for people, inspired by the recent works on affective vision and language including ArtEmis). We refer to reviewers to Figures 5 and 6 in the Appendix for example generation on Artemis 256.
> >
> >
> > >Table 2 shows DAMSM regularizer performs better than CLIP. It seems to be quite unusual since the representation power of DAMSM would be weaker than CLIP for most cases. Could you elaborate more why DAMSM is better than CLIP?
> >
> > Per the standard DAMSM practice, the instances of DAMSM model are pretrained on specific datasets such as MS-COCO, ArtEmis and CUB, separately. In contrast, CLIP is trained on a very huge dataset. Of course, the representation power of CLIP might be stronger but compared to DAMSM pretrained on specific dataset, it might not guide the model like DAMSM.
> >
> > >I’ve found that the comparison to SOTA in the T2I task is quite misleading, since the proposed method is not compared to real SOTA methods. In addition, the authors argue that the proposed method outperforms DALL-E 1, but DALL-E 1 reports the performance of zero-shot evaluation, which is not the case of this work.
> >
> > Thanks for your concerns. Let’s divide it into two parts.
> > >“Our method outperforms DALL-E1, but DALL-E1 results are based on zero-shot T2I generation.”
> >
> > We agree. We added a clarification in the paper to resolve this issue (see page 9 in the “Comparison to the State-of-the-Art” paragraph). We agree that even though our method outperforms DALL-E1 in image quality metrics, the comparison is not fair in one hand since DALL-E1 reports result in zero-shot T2I generation, but ours are based on standard T2I generation. DALL-E1 was trained on a large amount of data, orders of magnitudes larger than the benchmarks we are using and it may/may not cover data similar to these benchmarks. Since DALL-E1 is not released, we mainly meant to follow [1] on referring to the reported result mainly for reference.
> > [1] https://arxiv.org/pdf/2111.14822.pdf
> >
> > >“Comparison to SOTA is misleading since our method is not compared to real SOTA methods.”
> >
> > We respectfully disagree. XMC-GAN (CVPR2021)  (Table 5, row 4) and DF-GAN (CVPR2022)   (Table 5, row 5), are among the recently proposed SOTA methods in T2I task. Our aim is not to beat SOTA methods but rather to compare the proposed continuous method to existing methods and bring the gap closer. In addition, we are reporting FID scores of additional recent SSA-GAN [1] on MS-COCO and CUB benchmarks (results directly copied from the corresponding manuscripts):
> >
> > | method             | COCO  | CUB   |
> > |--------------------|-------|-------|
> > | SSA-GAN (CVRP2022) | 19.37 | 15.61 |
> > | Ours               | 18.58 | 11.00 |
> >
> > As you can observe, ours achieves better results than SSA-GAN on COCO and CUB.
> >
> >  [1]https://openaccess.thecvf.com/content/CVPR2022/papers/Liao_Text_to_Image_Generation_With_Semantic-Spatial_Aware_GAN_CVPR_2022_paper.pdf
> >
> > ---------------------------------------
> >
> > The concern below will be addressed in a follow-up response:
> >
> > >In my understanding, WhAtt does not take any sequence-level information. In CLIP, the sequence-level and sub-word level information are both useful representations, so it might be better to make use of both information in modulating the weights.

---

> > > ### Author Response · Authors · 2022-11-22
> > > **Part 3: Response to Reviewer cHep**
> > >
> > > >In my understanding, WhAtt does not take any sequence-level information. In CLIP, the sequence-level and sub-word level information are both useful representations, so it might be better to make use of both information in modulating the weights.
> > >
> > > Thanks for the suggestions. We hypothesize that sentence information is already captured with word embeddings in a way and in our current implementation of WhAtt attention enables the modulation to attend to multiple words, and hence capture the information in the entire sentence. However, other strategies can be also explored. We experimented with the following strategies that use both sentence and word embeddings. However, both strategies had a significantly slower convergence compared to our default version, and hence it was difficult to train.
> > >
> > > Concatenate sentence embedding with word embeddings before feeding them to hyper networks, and generate modulating tensor based on these concatenated features.  We hypothesize that the reason for slow convergence is that the model struggles to attend to relevant words due to extra sentence embedding.
> > >
> > > Pass sentence embeddings and word embeddings through their own separate hyper networks and generate modulating tensor M_s and M_w, respectively. Then, fuse these tensors into one M = M_s * M_w by pointwise multiplication. We hypothesize that the reason for slow convergence is that the modulating tensor from sentence embeddings M_s distorts/confuses the attention-based modulating tensor M_w from word embeddings.
> > >
> > >
> > > Another improvement that can be made ( orthogonal to WhAtt attention) is to add positioning embedding to the word embeddings to feed to WhAtt attention. This experiment is running and we will report it once the training is finished.

---

### Author Response · Authors · 2022-11-17
**Paper Revised Version**

We thank the reviewers for their valuable feedback. We are pleased that the reviewers find our method efficient in parameter size(Reviewer cHep, Reviewer 3nfM), We are glad that they found our approach to be interesting (Reviewer opUk), and novel (Reviewer 3nfM).  and achieving similar or better performance to other SOTA methods(Reviewer opUk, Reviewer xx7K). We address below the raised concerns and incorporate all feedback.
Key updates in the revised version are listed below including additional improvements requested by the reviewers.
1. Related work section has been updated to include a discussion about the diffusion models (Page 3).
2. Bottom Left picture of Figure 1 was updated to show more clear extrapolation behavior of the proposed model
3. Approach section was improved to increase clarity regarding
     1. Notations have been fixed in Section 3.2.1 (Page 5)
     2. Extreme tensor Factorization paragraph was enriched with an additional example (Page 6)
     3. Section 3.2.2, specifically WhAtt attention paragraph was revised to increase clarity regarding the information flow (Page 6)
     4. Some clarifying sentences were added about fairness in comparison to DALL-E in Page 9.
4. Some typos are fixed and some notations are fixed in an attempt to increase clarity in the paper
     1. Table 1 was updated to fix annotations.
     2. Table 9 was removed from the experiments section to Appendix
5. Code available:https://drive.google.com/file/d/1HDDlOll_0Vrd-y7vXj2-ph96jJkJ7K8t/view?usp=sharing
6. [project website] https://hypercgan-web.s3.amazonaws.com/index.html which has:
    1. code
    2. more qualitative examples

---

### Decision · Program_Chairs · 2023-01-20

**Decision:**

Reject

**Justification For Why Not Higher Score:**


There are several different concerns raised by the reviewers:

+ On the approach, the choices and details are sometimes unclear.
+ Mixed results and empirical analysis, including the ablation strategy and lack of details regarding parts of the setup.


**Justification For Why Not Lower Score:**

N/A

**Metareview: Summary, Strengths And Weaknesses:**

This work proposes text-to-image generation with a model, HyperCGAN, that is adversarially trained and leverages hypernetworks on top of both the generator and the discriminator. They also proposes an attention operator in order to encourage "grounding words to independent pixels at input (x, y) coordinates". They then demonstrate empirical results on several text-image datasets.

There are several different concerns raised by the reviewers:

+ On the approach, the choices and details are sometimes unclear.
+ Mixed results and empirical analysis, including the ablation strategy and lack of details regarding parts of the setup.

Reviewers generally leaned toward reject. The one exception was one reviewer who did not provide many details and unfortunately did not participate during discussion. I agree with the majority consensus.